# *Sinorhizobium meliloti* DnaJ Is Required for Surface Motility, Stress Tolerance, and for Efficient Nodulation and Symbiotic Nitrogen Fixation

**DOI:** 10.3390/ijms24065848

**Published:** 2023-03-19

**Authors:** Paula Brito-Santana, Julián J. Duque-Pedraza, Lydia M. Bernabéu-Roda, Cristina Carvia-Hermoso, Virginia Cuéllar, Francisco Fuentes-Romero, Sebastián Acosta-Jurado, José-María Vinardell, María J. Soto

**Affiliations:** 1Estación Experimental del Zaidín, CSIC, Department of Biotechnology and Environmental Protection, 18008 Granada, Spain; 2Facultad de Biología, Departamento de Microbiología, Universidad de Sevilla, 41012 Sevilla, Spain; 3Centro Andaluz de Biología del Desarrollo, CSIC, Junta de Andalucía, Departamento de Biología Molecular e Ingeniería Bioquímica, Universidad Pablo de Olavide, 41013 Seville, Spain

**Keywords:** Rhizobium, plant colonization, surface motility, flagella, chaperone, stress tolerance, nodulation, nitrogen fixation

## Abstract

Bacterial surface motility is a complex microbial trait that contributes to host colonization. However, the knowledge about regulatory mechanisms that control surface translocation in rhizobia and their role in the establishment of symbiosis with legumes is still limited. Recently, 2-tridecanone (2-TDC) was identified as an infochemical in bacteria that hampers microbial colonization of plants. In the alfalfa symbiont *Sinorhizobium meliloti*, 2-TDC promotes a mode of surface motility that is mostly independent of flagella. To understand the mechanism of action of 2-TDC in *S. meliloti* and unveil genes putatively involved in plant colonization, Tn*5* transposants derived from a flagellaless strain that were impaired in 2-TDC-induced surface spreading were isolated and genetically characterized. In one of the mutants, the gene coding for the chaperone DnaJ was inactivated. Characterization of this transposant and newly obtained flagella-minus and flagella-plus *dnaJ* deletion mutants revealed that DnaJ is essential for surface translocation, while it plays a minor role in swimming motility. DnaJ loss-of-function reduces salt and oxidative stress tolerance in *S. meliloti* and hinders the establishment of efficient symbiosis by affecting nodule formation efficiency, cellular infection, and nitrogen fixation. Intriguingly, the lack of DnaJ causes more severe defects in a flagellaless background. This work highlights the role of DnaJ in the free-living and symbiotic lifestyles of *S. meliloti*.

## 1. Introduction

Rhizobia are soil-dwelling alpha- and betaproteobacteria, which are able to establish nitrogen-fixing symbiosis with legumes [1]. In nitrogen-poor soils, these bacteria can elicit the formation of new organs, known as nodules, on the roots of their host plants. Root nodules are invaded by rhizobia where they differentiate into bacteroids capable of reducing atmospheric dinitrogen into ammonia that is provided to the plant, relieving its dependency on nitrogen fertilizers. Bacteria also benefit from this symbiosis by receiving a carbon source and essential nutrients from the plant, together with a protected environmental niche [2,3].

The development of nitrogen-fixing nodules in legume roots is the outcome of a complex process that involves a highly specific and continuous molecular dialogue between the two symbiotic partners, and of which much knowledge has been gained in the last years [1,4,5,6]. An early event crucial for the establishment of symbiosis is bacterial root colonization. This process involves several steps beginning with the directed movement of bacteria toward the roots, followed by the attachment of bacteria to the root surface (i.e., rhizoplane) and subsequent formation of a biofilm in which cells are embedded in an extracellular matrix that confers protection against adverse conditions [7,8]. In contrast to the well-known processes of nodule formation and nitrogen fixation, the knowledge about mechanisms used by rhizobia for rhizoplane colonization is scarce. Recently, a few genome-wide approaches have successfully been adopted to gain insights into this field and motility has been found to play a fundamental role [9,10,11]. Chemotaxis and motility are not essential for nodulation or nitrogen fixation but are crucial for competitive root colonization, which in turn can influence nodule formation efficiency and competitiveness [12,13,14]. Flagella-driven swimming motility permits individual bacteria to move in aqueous environments and allows rhizobia to approach its host plant in response to chemoattractants released by legume roots, as well as to find proper sites for infection [13,14].

In nature, microbes are usually associated with biotic and abiotic surfaces. To be able to colonize these niches, bacteria have evolved different motility strategies [15,16,17,18]. Swarming is probably the most extensively studied type of surface motility because it is a trait closely connected with biofilm formation and virulence in pathogenic bacteria [19,20]. Swarming is a flagella-driven motility characterized by the rapid coordinated multicellular migration of bacteria across solid surfaces [19]. Bacteria can also move over surfaces using sliding, a passive appendage-independent spreading in which surfactants and other compounds that diminish friction between cells and surfaces play a fundamental role [21]. In contrast to swimming, during surface motility, bacteria face important challenges that need to be overcome. Among them, bacteria need to attract water to the surface to allow for flagellar activity, overcome frictional forces, and reduce surface tension [17,21,22]. Several physical, chemical, and biological factors are known to influence the bacterial capacity to translocate across surfaces, which convert surface motility into a complex and highly regulated phenotype [17,18]. Rhizobia can move over surfaces by using flagella-dependent and independent mechanisms [23,24,25,26,27,28,29]. However, knowledge about the regulatory mechanisms that control surface motility in rhizobia as well as the role of this bacterial trait in the establishment of symbiosis is limited [28,29,30,31].

*Sinorhizobium meliloti*, the alfalfa endosymbiont, can translocate over semisolid surfaces by using flagella-mediated mechanisms, or by sliding promoted by the production of exopolysaccharides and surfactants [23,28,32,33,34,35]. Like in other bacteria, studies on the mechanisms underlying surface motility in *S. meliloti* revealed a close connection with biofilm formation as these two surface-associated traits were influenced by the same regulatory pathways and chemical cues [28,29,36]. The characterization of an *S. meliloti* mutant that exhibits increased surface motility, defects in biofilm formation, and impaired root colonization, led to the discovery of 2-tridecanone (2-TDC) as an infochemical that affects bacterial surface-associated traits and hampers microbial colonization of plant tissues [36]. 2-TDC is a volatile methylketone known as a natural insecticide produced in high amounts by wild tomato plants [37]. Several rhizobacteria, including *S. meliloti*, can also produce 2-TDC [36,38,39]. Our investigations revealed that, without affecting bacterial growth or swimming motility, exogenous application of 2-TDC promoted surface motility and impaired biofilm formation in *S. meliloti* [36]. Moreover, the presence of 2-TDC impairs the nodulation of alfalfa by hampering the bacterial ability to efficiently colonize plant roots. Therefore, the mechanistic understanding of the effects caused by 2-TDC on *S. meliloti* could potentially unveil bacterial genes required for symbiosis.

In *S. meliloti* strain GR4 [40], 2-TDC promotes a mode of surface motility that is mostly independent of flagella and mediated by an as-yet-unknown mechanism [36]. To decipher the molecular bases responsible for the volatile-triggered surface spreading, transposants derived from the flagellaless GR4flaAB strain exhibiting limited or no surface motility in the presence of 2-TDC were isolated and genetically characterized. In one of these mutants, the transposon interrupted the *dnaJ* gene, which potentially codes for a molecular chaperone homologous to the eukaryotic 40 kDa heat shock protein Hsp40. The DnaJ protein belongs to the ATP-dependent chaperone folding system DnaK/DnaJ/GrpE [41]. DnaJ functions as a chaperone and holdase that transfers unfolded, misfolded, or aggregated proteins to DnaK, which is responsible for the folding of the substrate with the participation of the nucleotide exchange factor GrpE. With less efficiency than DnaK, DnaJ can also work autonomously as a foldase of denatured proteins [41,42,43]. The lack of DnaJ has been associated in some bacteria with the inability to produce flagella or impaired swimming and swarming motilities [44,45,46]. Moreover, the participation of bacterial DnaJ in biofilm formation, the colonization of eukaryotic hosts, and pathogenicity has been shown [47,48,49,50,51].

The role of DnaJ in legume endosymbionts has scarcely been investigated. Interestingly, the lack of this chaperone affects differently the establishment of symbiosis depending on the rhizobial species [52,53,54]. In this work, the role of *S. meliloti* DnaJ in the bacterial response to volatile 2-TDC, as well as in the adaptation to stress conditions and the establishment of symbiosis with alfalfa plants, has been investigated. Our results show that DnaJ is essential in the flagella-independent surface motility triggered by volatile 2-TDC and in the swarming motility exhibited by the wild-type *S. meliloti* strain GR4, but it plays a minor role in swimming motility. In addition, DnaJ helps *S. meliloti* cells to adapt to stressful conditions and is required for efficient nodule colonization and symbiotic nitrogen fixation in alfalfa plants.

## 2. Results

### 2.1. Isolation and Genetic Characterization of Flagellaless GR4flaAB-Derivative Mutants That Do Not Respond to Volatile 2-TDC

To identify *S. meliloti* genes playing a role in the flagella-independent motility triggered by volatile 2-TDC, strain GR4flaAB was subject to Tn*5* transposon mutagenesis and the resulting kanamycin-resistant transposants were tested for surface motility on 1% agar minimal medium (MM) in the presence of volatile 2-TDC. After screening 3.885 transposants, five mutants named NS (for nonspreading) were identified as impaired in the response to the volatile, i.e., they exhibited reduced or no translocation across the surface in the presence of airborne 2-TDC in contrast to the parental strain GR4flaAB (Appendix A). To determine the genomic location of each insertion, arbitrary PCR and DNA sequencing were performed (see Section 4). Genetic characterization of the mutants revealed five genes that could potentially be involved in the action mechanism of 2-TDC (Table 1).

Only one of the five genes identified in our screening, the *actS* gene, was previously characterized in *S. meliloti*. The *actS* gene encodes the sensor histidine kinase of the two-component regulatory system ActS/ActR that has been associated with bacterial tolerance to acidic pH, the adaptation to oxidative stress, the regulation of microaerobic respiration, and cell envelope function [55,56,57,58]. Transposon insertions in mutants NS2, NS3, and NS4 were located in the *cmk*, *ctaA*, and *dnaJ* genes that code for proteins involved in the phosphorylation of (d)CMP, heme A synthesis, and in chaperoning/protein homeostasis, respectively. Finally, in the NS5 mutant, the transposon was located on a chromosomal gene that in GR4 codes for a protein putatively involved in the metabolism of the second messenger cyclic-dimeric guanosine monophosphate (c-di-GMP). This protein is not produced in the well-known strains Rm1021/Rm2011 due to the truncation of *smc03142* by a non-sense mutation [59]. In this study, we focused on the role of *dnaJ*, the gene inactivated by Tn*5* in NS4, in the response of *S. meliloti* to 2-TDC and its participation in plant colonization.

### 2.2. In Silico Analyses of S. meliloti dnaJ

The C770_GR4Chr0186 locus affected in the NS4 transposant is orthologous to the *smc02858* gene of Rm1021 that is annotated as *dnaJ.* In the reference Rm1021 strain, *dnaJ* is flanked upstream by the chaperone Hsp70-encoding *dnaK* gene and downstream by the *smc02859*-*smc02860* genes that are transcribed in the opposite direction to *dnaJ* and encode proteins of unknown function. The genomic context of *dnaJ* in strain GR4 is similar to that of Rm1021 except that a copy of the insertion sequence IS110 is located between *dnaJ* and the *smc02859*-*smc02860* operon (Figure 1a). In contrast to the genetic organization found in many bacteria [42], in *S. meliloti*, *grpE* (*smc01142*) is not in the proximity of the *dnaK dnaJ* genes.

In GR4, the *dnaJ* gene encodes a 379 amino acid protein, which is 100% identical to the protein of Rm1021 and shows a domain architecture typical of class A JDP members (Figure 1b) [42]. The amino terminal J-domain, which is essential for functional interaction with DnaK, contains the invariant tripeptide of histidine, proline, and aspartic acid (HPD motif) [42,43]. Next to the N-terminal J-domain, a glycine/phenylalanine (G/F)-rich region connects the J-domain to a zinc-binding domain (ZBD), which is followed by a C-terminal domain (CTD). The ZBD is a cysteine-rich region with four CXXCXGXG (C, cysteine; G, glycine; X, any amino acid) repeats that coordinate two zinc ions. This domain participates in substrate binding and activation of the DnaK chaperone and also harbors oxidoreductase activity [42,49,60]. The CTD is the main substrate binding domain of DnaJ, although the (G/F)-rich region and the ZBD also contribute to this role [42,43]. In the NS4 mutant, the Tn*5* insertion is located in the ZBD of DnaJ affecting residue D_201_, which could lead to a truncated DnaJ protein.

The DnaJ protein of *S. meliloti* shows high identities with DnaJ proteins from different rhizobia such as *Sinorhizobium medicae* WSM419 (96.6%), *Rhizobium etli* CFN42 (87.2%), *R. tropici* CIAT899 (87%), *Rhizobium leguminosarum* biovar *viciae* 3841 (86.6%), and *B. diazoefficiens* USDA 110 (66.1%). It also shows high identities with DnaJ proteins of different pathogenic and beneficial bacteria, such as *Agrobacterium tumefaciens* (85.3%), *Brucella ovis* (78.9%), *Salmonella enterica* subsp. *enterica* (75.7%), *Escherichia coli* K12 (55.2%), or *Pseudomonas putida* (54.7%). Multiple sequence alignments of these proteins reveal that the different domains and motifs characteristic of DnaJ are conserved, although the (G/F)-rich region exhibits some variability (Appendix A).

### 2.3. DnaJ Is Required for Flagella-Dependent and Independent Types of Surface Motility but It Plays a Minor Role in Swimming

To confirm that the inactivation of the *dnaJ* gene caused by the Tn*5* insertion in the NS4 transposant was responsible for the 2-TDC insensitivity of the mutant, complementation experiments were performed using a plasmid construct that expresses the wild-type *dnaJ* allele. As shown in Figure 2, the ectopic expression of *dnaJ* in NS4 (NS4 C) recovered the ability of the mutant to spread over the surface of the medium in the presence of volatile 2-TDC at levels similar to those exhibited by the parental strain GR4flaAB carrying the empty vector (flaAB ev).

We also constructed in-frame markerless *dnaJ* deletion mutants derived from the flagellaless strain GR4flaAB and the wild-type strain GR4 (flaABΔdnaJ and GΔdnaJ strains, respectively). No major differences in growth were detected between the *dnaJ* mutants and their parental strains when serial dilutions of cultures grown to the mid-exponential phase were spotted onto MM plates (Appendix A). On TY plates, a difference in colony size was observed between the mutants and the parental strains, with the *dnaJ* mutants developing smaller colonies (Appendix A). Growth curves performed in liquid MM and TY revealed slower growth of the *dnaJ* mutants compared to their parental strains (Appendix A). The retarded growth shown by the mutants was mainly due to their longer lag phases, which suggests that the inactivation of *dnaJ* impairs the adjustment of cells to new environmental conditions. The most affected strain during growth in liquid media was flaABΔdnaJ, whereas GΔdnaJ and NS4 exhibited similar behavior to each other. The differences in growth observed between flaABΔdnaJ and GΔdnaJ indicate that the role of DnaJ is more relevant for the flagellaless strain than for the wild-type strain. Moreover, the faster growth exhibited by NS4 compared to the deletion mutant flaABΔdnaJ suggests the existence in NS4 of some DnaJ-related activity.

The deletion mutants flaABΔdnaJ and GΔdnaJ were used in surface motility assays in the presence of volatile 2-TDC. Like NS4, neither of the two *dnaJ* deletion mutants was able to spread over the surface of the medium in the presence of the volatile (Figure 3a,b). The fact that GΔdnaJ had the same phenotype as flaABΔdnaJ in the presence of airborne 2-TDC was not unexpected since 2-TDC stimulates a mode of surface motility that is mostly independent of flagella. Surface translocation in response to 2-TDC was restored in the two deletion mutants carrying the *dnaJ*-complementing plasmid (Figure 3a,b), clearly demonstrating that DnaJ plays an essential role in the flagella-independent surface motility promoted by 2-TDC in *S. meliloti*.

Swimming and swarming motilities exhibited by GR4 are abolished in GR4flaAB because of its inability to produce flagella [30]. Surprisingly, the lack of DnaJ also abolished the flagella-dependent swarming motility exhibited by the wild-type strain GR4 under permissive conditions (MM 0.6% agar) (Figure 4a), indicating that DnaJ is not only required for the bacterial response to 2-TDC but seems to have a broader role in surface translocation in *S. meliloti*. Interestingly, the deletion of *dnaJ* in GR4 only had a minor impact on swimming motility with a slight reduction in the swimming halo compared with that of the wild-type strain (10 ± 0.02 mm vs. 14 ± 0.02 mm; *p* < 0.001) (Figure 4b). Compatible with the results obtained in swimming, transmission electron microscopy observations revealed that the GR4-derived *dnaJ* mutant produced flagella, which were indistinguishable from those produced by GR4 (Figure 4c). These results indicate that the lack of DnaJ in *S. meliloti* does not abolish flagella production or function. The requirement of DnaJ for surface motility, which contrasts with its minor role for swimming, suggests that the chaperone participates in specific mechanisms involved in bacterial translocation across surfaces.

### 2.4. DnaJ Participates in Salt Stress Tolerance in S. meliloti

In *Rhizobium tropici*, *dnaJ* has been involved in salt tolerance [54]. To determine if *dnaJ* in *S. meliloti* plays a similar role, the growth of the *dnaJ* deletion mutant strains and the NS4 transposant was tested on MM supplemented with NaCl (300 mM) and compared to that of their parental strains (Figure 5a). In the absence of salt stress (Control), no differences in growth were observed among the different strains. In the NaCl-supplemented medium, the parental strains GR4 and GR4flaAB exhibited a slightly slower growth compared with the control medium, an effect that was noticeable after 5 days of incubation. When the plates were incubated for longer (17 days), no decrease in GR4 and GR4flaAB cell survival was observed in response to salt stress since a similar number of colony-forming units were detected in media with or without the stressor. In contrast, salt stress negatively impacted the growth of the *dnaJ* mutants, with differences depending on the strain (Figure 5a). Growth was especially impaired in the flaABΔdnaJ mutant, in which almost no growth could be detected after 5 days of incubation. Interestingly, the delay in growth of the NS4 transposant was less severe than for flaABΔdnaJ, indicating that the deletion of *dnaJ* in the GR4flaAB strain has a stronger impact on salt tolerance than the insertional inactivation of the gene. The GR4-derived *dnaJ* deletion mutant also showed slower growth than the parental strain in the presence of NaCl, but it was clearly less affected than the GR4flaAB-derived deletion mutant (flaABΔdnaJ). The difference between GΔdnaJ and flaABΔdnaJ was even more obvious when the NaCl-supplemented plates were incubated for longer. Whereas GΔdnaJ hardly showed any decrease in cell survival, in flaABΔdnaJ, the number of cells was reduced by approx. 2–3 orders of magnitude compared to GΔdnaJ or NS4 (Figure 5a).

To test whether the salt-sensitive phenotype of the mutants was due to the inactivation of *dnaJ*, complementation experiments were performed (Figure 5b). Under control conditions and after 5 days of incubation, no significant differences in growth were detected among the mutants, regardless of whether they were carrying the empty vector pJB3 (ev) or the *dnaJ*-expressing plasmid construct (C). With the same incubation times, salt stress notably reduced the growth of all strains but with a stronger effect in those harboring the empty vector (Figure 5b). A longer incubation revealed that salt stress not only reduced growth rate but also cell survival in the *dnaJ* mutants carrying the empty vector. Interestingly, the number of *dnaJ* mutant cells that survived during growth under salt stress conditions was lower for strains carrying the empty vector pJB3 and grown on media supplemented with antibiotics than for the same strains without pJB3 (Figure 5a,b). Most likely, the tetracycline added to the plates to ensure plasmid maintenance and/or the burden of carrying additional DNA represent added stresses for which cells might also require the DnaJ function. These conditions also revealed differences in cell survival between GΔdnaJ and NS4 that were not detected when the same strains without plasmid pJB3 were grown in media without antibiotics. The least sensitive strain was GΔdnaJ followed by NS4. Strain flaABΔdnaJ was the most sensitive with hardly any growth after 17 days of incubation. The salt-sensitive phenotype exhibited by the three mutants was recovered with the ectopic expression of *dnaJ* from the plasmid construct pJ-dnaJ (C), a complementation effect that also abolished differences between the mutants (Figure 5b). Altogether, these results demonstrate that DnaJ contributes to *S. meliloti* adaptation to salt stress. Our data also indicate that the role of DnaJ in managing salt stress is more important for the flagellaless strain GR4flaAB than for GR4. Moreover, the increased salt tolerance shown by NS4 compared to the deletion mutant flaABΔdnaJ suggests that the putative DnaJ truncated protein likely produced by the transposant might retain some activity that helps to cope with salt stress.

### 2.5. DnaJ Protects S. meliloti against Oxidative Stress

The DnaK/DnaJ chaperone protects *Salmonella* against oxidative stress created by reactive oxygen species that are produced by phagocytes in the innate host response [61]. In this study, we investigated whether DnaJ plays a role in the protection of *S. meliloti* against oxidative stress. For that, the growth of the mutants and their parental strains was tested on MM supplemented with different concentrations of hydrogen peroxide (H_2_O_2_). When 100 µM H_2_O_2_ was added to the medium, a slight delay in growth was observed for the three *dnaJ* mutants compared with the parental strains. Under these conditions, flaABΔdnaJ was the most affected mutant, whereas NS4 and GΔdnaJ exhibited similar growth (Figure 6a). No effects on cell survival were observed for any of the strains under these conditions and, with longer incubation time, the growth of the mutants was indistinguishable from that of the parental strains.

When the concentration of H_2_O_2_ was increased up to 150 µM, cell survival was significantly reduced in all the strains by three to five orders of magnitude. The decrease in cell survival was stronger for the *dnaJ* mutants than for their parental strains, indicating that DnaJ protects *S. meliloti* against oxidative stress. As observed for osmotic stress, the *dnaJ* mutants derived from GR4flaAB were more sensitive than the GR4-derived deletion mutant, highlighting the important role of DnaJ in the flagellaless strain. Interestingly, we found that GR4flaAB was slightly more tolerant to oxidative stress than wild-type GR4 (Figure 6a). This result was confirmed by determining the number of GR4 and GR4flaAB cells that survived after exposure to 1 mM H_2_O_2_ for 90 min (Appendix A). In two independent experiments, the percent of GR4flaAB cells that survived the stress was 2.3- and 3-fold higher than in GR4 (*p* < 0.0003), demonstrating the modest but significantly higher tolerance to oxidative stress of the flagellaless strain.

Complementation experiments were also performed under these stress conditions (Figure 6b). Similarly, as was observed in response to salt stress (Figure 5b), with the same concentration of stressor (100 µM H_2_O_2_), the growth of strains carrying the empty vector on media with an antibiotic was much more severely affected than for the same strains without plasmids, indicating greater bacterial stress. The ectopic expression of *dnaJ* improved the tolerance of flaABΔdnaJ to H_2_O_2_ slightly, but not in NS4 or GΔdnaJ. This could indicate that the oxidative stress conditions used in our experiments are interfering somehow with the DnaJ activity.

### 2.6. S. meliloti DnaJ Is Required for Nodule Formation Efficiency and Symbiotic Nitrogen Fixation in Alfalfa Plants

The requirement of DnaJ for the establishment of efficient nitrogen-fixing symbiosis with legumes depends on the rhizobial species. To investigate the contribution of *S. meliloti* DnaJ in symbiosis, the three *dnaJ* mutants obtained in this study were used to inoculate alfalfa plants, and their symbiotic performance was compared to that of their parental strains GR4 and GR4flaAB. Nodulation kinetics experiments revealed that the *dnaJ* mutants were able to induce the formation of nodules on alfalfa roots but with lower nodule formation efficiencies than their parental strains (Figure 7a,b). Thus, compared with the wild-type strain GR4, the deletion mutant GΔdnaJ exhibited a delay in the appearance of nodules and a reduction in the number of nodules developed per plant during the first ten days after inoculation (Figure 7a). At the end of the experiment, no significant differences were detected in the number of nodules induced by the wt or the mutant strain. Similar behavior was observed for the deletion mutant flaABΔdnaJ and the NS4 transposant when their nodulation kinetics were compared to that of their parental strain GR4flaAB (Figure 7b). Notably, the delay in nodulation was more severe in plants inoculated with flaABΔdnaJ than in those inoculated with NS4 or GΔdnaJ. These results indicate that DnaJ is important for nodule formation efficiency in *S. meliloti*, especially in a flagella-minus strain. Moreover, the difference exhibited in the nodulation kinetics between flaABΔdnaJ and NS4 suggests that the latter maintains some DnaJ-related activity that assists during the nodulation process.

In addition to showing delayed nodulation, alfalfa plants inoculated with either of the *dnaJ* mutants exhibited symptoms of nitrogen deficiency at the end of the experiment (three weeks after inoculation), as shown by reduced plant growth and the chlorotic appearance of the plant shoots (Appendix A). Compatible with these observations, the shoot dry weights of plants inoculated with the mutants were significantly reduced compared to those inoculated with the wild-type and flaAB strains (Figure 7c). Impaired plant growth was more obvious in plants inoculated with the *dnaJ* deletion mutants than in those inoculated with NS4. Importantly, all three *dnaJ* mutants recovered the nodule formation efficiency of their parental strains upon the expression of *dnaJ* from a plasmid construct (Appendix A). Inoculation with the complemented strains also reversed the symptoms of nitrogen deficiency caused by *dnaJ* loss-of-function (Appendix A).

To determine if the impaired symbiotic effectiveness of the *dnaJ* mutants was caused by a defect in nodule invasion and/or nitrogenase activity, nodulation experiments were performed in Leonard jars. Thirty days after inoculation, plants treated with the *dnaJ* mutants showed clear symptoms of nitrogen deficiency (Figure 8). As already observed in the nodulation kinetics (Figure 7a,b), all five *S. meliloti* strains were able to elicit the formation of nodules. However, whereas nodules elicited by the parental strains showed a uniform strong pink color characteristic of the symbiotic leghemoglobin, nodules induced by the *dnaJ* mutants were smaller, with a lighter pink color in the apical part and a greenish color in the proximal part indicating senescence (leghemoglobin degradation) (Figure 8).

Shoot dry weight (SDW) analyses revealed that inoculation with any of the five *S. meliloti* strains used in this study significantly increased plant growth with respect to noninoculated plants (*p* < 0.01 for flaABΔdnaJ; *p* < 0.001 for the remaining four strains) (Table 2). As already reported [30], alfalfa plants inoculated with the flagellaless strain GR4flaAB showed reduced shoot dry weight compared to plants inoculated with the wild-type strain (*p* < 0.001), which correlates with reductions in the number of nodules (*p* < 0.05) and nodule fresh weight (*p* < 0.01). Consistent with data presented in Figure 7c, plants inoculated with any of the three *dnaJ* mutants exhibited a significant reduction in plant growth compared to plants inoculated with the corresponding parental strain (Table 2). This phenotype correlated with significant reductions in nodule fresh weight (NFW) and acetylene reduction activity (ARA), but not with reductions in the number of nodules (NN). In fact, plants inoculated with NS4 elicited a slightly higher number of nodules than the parental GR4flaAB strain.

To understand the symbiotic phenotype shown by the *dnaJ* mutants, nodules induced by the different strains were analyzed by light microscopy (Figure 9).

The results revealed that the *dnaJ* mutant strains were able to colonize the nodule cells and differentiate into bacteroids. However, nodules elicited by the mutants appeared to contain fewer infected cells than the nodules induced by the parental strains (Figure 9). To confirm this observation, the number of infected and noninfected cells was determined. The percent of noninfected cells in sections of nodules induced by GR4 and GR4flaAB was similar (14.6% ± 3.7% vs. 17.6% ± 3.3%; *p* = 0.6). In contrast, the number of noninfected cells in nodules induced by the GR4 derivative *dnaJ* mutant was 2.1 times higher than that of GR4 (30.7% ± 1.9% vs. 14.6% ± 3.7%; *p* = 0.005). Notably, nodules induced by flaABΔdnaJ contained the highest percent of noninfected cells, which was 2.8 times higher than that of its parental strain (48.6% ± 1.8% vs. 17.6% ± 3.3%; *p* = 0.0001). The NS4 transposant showed an intermediate phenotype between the parental flaAB strain and the deletion mutant, with a percent of noninfected cells 2.2 times higher than in nodules induced by the flaAB strain (39.1% ± 3.7% vs. 17.6% ± 3.3%; *p* = 0.008). Moreover, significant starch accumulation was observed in nodules induced by flaABΔdnaJ and NS4, a feature that is commonly associated with reduced or abolished nitrogen-fixing activity [62]. These results indicate the need for DnaJ for efficient nodule colonization and symbiotic nitrogen fixation.

## 3. Discussion

The characterization of the molecular bases underlying surface motility in *S. meliloti* unveiled the importance of the chaperone DnaJ in different aspects of the free-living and symbiotic lifestyles of this alfalfa symbiont. Ex planta, we found that DnaJ is essential for bacterial surface motility but not for swimming. Notably, DnaJ contributes to bacterial endurance under salt and oxidative stress, especially in a flagellaless background. In symbiosis with alfalfa, the lack of DnaJ impairs nodule formation efficiency, nodule colonization, and, consequently, symbiotic nitrogen fixation. The lack of flagella increases the symbiotic defects of a *dnaJ* mutant, consistent with the higher sensitivity to stress shown by this strain.

We found the *dnaJ* gene as the locus inactivated in NS4, a mutant unable to exhibit the flagella-independent surface motility triggered by volatile 2-TDC. This compound functions as an infochemical in bacteria [36]. In *S. meliloti,* 2-TDC promotes bacterial surface motility, reduces biofilm formation on abiotic surfaces and on plant roots, and impairs the nodulation of alfalfa [36]. In general, very little is known about how bacteria perceive and respond to microbial volatiles [63]. With this study, we aimed at gaining insights into this field and specifically about the molecular bases responsible for the flagella-independent surface translocation triggered by 2-TDC in *S. meliloti*. Moreover, considering the fundamental role of motility and biofilm formation on plant root colonization, we reasoned that by characterizing the action mechanism of 2-TDC in *S. meliloti*, we could unveil bacterial genes and mechanisms relevant during the early stages of the symbiosis. Intriguingly, out of the five genes identified in our screening, two of them (*actS* and *dnaJ*) are involved in bacterial adaptation to stress, while the remaining three (*cmk*, *ctaA*, and C770_GR4Chr3081) await further investigation. ActS is required for tolerance to acidic pH and adaptation to oxidative stress [55,58], and data obtained in the present study demonstrate that DnaJ participates in tolerance to osmotic stress and, to a lesser extent, oxidative stress. Previous transcriptome analyses performed in *S. meliloti* to identify 2-TDC-regulated functions revealed the upregulation of some genes related to stress responses [36]. This, together with the results obtained in our genetic screen, suggest that 2-TDC is triggering some stress-related response that does not affect bacterial growth but stimulates surface motility. The positive correlation between stress and activation of surface motility has already been reported in the literature. In *Sinorhizobium fredii* HH103, osmotic nonionic stress activates swarming while it abolishes swimming motility [64], and in *Pseudomonas aeruginosa*, the stringent stress response is essential for surfing motility, a specialized type of surface translocation that is induced by mucin, the viscous mucus produced by cystic fibrosis-affected lungs [65]. It is tempting to speculate that the swarming motility exhibited by the wild-type GR4 in the absence of 2-TDC is also the result of bacterial adaptation to stressful conditions, in which DnaJ plays a pivotal role.

The DnaJ protein belongs to the ATP-dependent chaperone folding system DnaK/DnaJ/GrpE and plays a fundamental role in microbial proteostasis under both normal and stress conditions [41,42,43]. The gene altered in mutant NS4 codes for a protein that shows the characteristic domain architecture of class A J-domain proteins that function as co-chaperones of DnaK [42] and exhibits high identities with DnaJ proteins from different bacteria. These data strongly suggest that the gene interrupted by the Tn*5* insertion in NS4 codes for the DnaJ chaperone of *S. meliloti*. In bacteria, the *dnaK*, *dnaJ*, and *grpE* genes are often found within the same operon, which is constitutively expressed under normal conditions and becomes upregulated in response to heat stress [41,42]. In *S. meliloti, grpE* is not linked to the *dnaKdnaJ* operon, which is known to be regulated by the alternative σ factor RpoH1 under heat stress conditions [66]. The identification of an antisense RNA (SMc_asRNA_774) that overlaps the 5′-UTR of the mRNA initiated upstream of *dnaK*, as well as additional transcriptional start sites mapped immediately upstream of *dnaJ* [67], suggest a more complex regulation of these genes that warrant investigation.

Like the NS4 transposant, newly constructed in-frame markerless *dnaJ* deletion mutants derived from the flagellaless GR4flaAB and the wild-type GR4 strains were unable to translocate over surfaces in response to volatile 2-TDC. The nonmotile phenotype of all three mutants was restored with the ectopic expression of the wild-type *dnaJ* allele, clearly demonstrating the involvement of DnaJ in the flagella-independent surface motility triggered by 2-TDC in *S. meliloti*. Interestingly, we also found that DnaJ loss-of-function abolished the flagella-driven swarming motility exhibited by the wild-type strain GR4. This result indicates that DnaJ is not only required for the bacterial response to 2-TDC but seems to have a broader role in surface translocation in *S. meliloti*. It is known that *Escherichia coli dnaJ* mutants lack flagella and are nonmotile [44], and the deletion of *dnaJ* in *Pseudomonas putida* strains reduces swimming and the production of the swarming-stimulating biosurfactant putisolvin [45,46]. Impaired flagella production or activity in the GR4-derived *dnaJ* mutant could explain its inability to swarm. However, in contrast to *E. coli*, the GR4-derived *dnaJ* mutant produced flagella and exhibited swimming motility, which was only slightly reduced compared to that of the wild-type strain. Hence, the requirement of DnaJ for surface motility but its minor role for swimming suggests that the chaperone participates in specific mechanisms for bacterial translocation across surfaces, yet to be discovered.

DnaJ has been involved in stress tolerance in different bacteria [49,52,54,68]. Results obtained in this study show that *S. meliloti* DnaJ participates in salt tolerance and, to a lesser extent, in adaptation to oxidative stress. Interestingly, we found that under the same stress conditions, the role of DnaJ was more relevant for the flagellaless GR4flaAB strain than for the wild-type. Thus, under salt stress conditions, the deletion of *dnaJ* in GR4 led to slower growth compared to the wild-type strain, whereas the same mutation in GR4flaAB not only slowed growth but also led to a significant reduction in cell survival (Figure 5a). Likewise, the slower growth and reduced cell survival caused by H_2_O_2_ in the two *dnaJ* deletion mutants were more pronounced in the flagellaless background. The increased sensitivity to stress shown by the flagella-minus flaABΔdnaJ mutant compared to the flagella-plus ΔdnaJ mutant suggests that, under the same conditions, the flagellaless strain is experiencing stronger stress. However, when harboring a functional *dnaJ* gene, GR4flaAB did not show symptoms of lower tolerance than GR4 to salt or oxidative stress. In fact, GR4flaAB was slightly more tolerant to oxidative stress than GR4. Improved tolerance to oxidative stress of nonflagellated bacteria has already been described for *P. putida*. This behavior was explained by the increased energy (ATP) and reducing power (NADPH) resulting from avoiding the metabolic costs of the production and rotation of flagella [69]. GR4flaAB is unable to produce functional flagella because it lacks the principal flagellin subunit FlaA as well as FlaB, one of the other three accessory subunits required to form the complex *S. meliloti* flagellar filament [70]. Whether the lack of flagellar filaments and flagellar rotation and/or the production of incomplete flagellar structures hamper the proper activation of stress responses in the cell upon exposure to an insult, which would make DnaJ activity more vital for endurance, warrants further investigation.

When testing tolerance to osmotic stress, it was noticed that the flagellaless *dnaJ* insertion mutant NS4 was clearly more tolerant than flaABΔdnaJ, suggesting the existence of some DnaJ activity in the transposant. In NS4, the Tn*5* insertion was located downstream of the sequence coding for the third CXXCXGXG repeat of the ZBD of DnaJ (Figure 1b). Therefore, the possibility exists that a truncated form of DnaJ comprising the J-domain, the G/F rich region, and at least one zinc finger, is being produced in NS4. In *E. coli*, protein refolding requires the entire DnaJ protein, but the J- and G/F domains are known to be sufficient for the interaction with DnaK and the stimulation of its ATPase activity, which is mandatory for the folding of the substrate [42,43]. The existence of this partial activity of DnaJ in NS4 could explain why this transposant exhibits better performance under osmotic stress conditions compared to the deletion mutant. However, the possibility that the greater tolerance exhibited by NS4 is the result of loss of transposon or the selection of suppressor mutations cannot be ruled out.

Finally, results obtained in this work demonstrate that *S. meliloti* DnaJ plays an important role in the establishment of efficient symbiosis with alfalfa plants. Our data indicate that this chaperone is required at early and late stages of symbiosis development, such as nodule formation efficiency and nodule colonization. This is in contrast with the phenotype exhibited by a *B. japonicum dnaJ* mutant, which can establish efficient symbiosis with soybean [52], but in line with *dnaJ* mutants in *R. tropici* and *R. leguminosarum* bv. *phaseoli*, which form ineffective nodules with beans [53,54]. The symbiotic performance exhibited by our three *dnaJ* mutants correlates with their tolerance to stress, with the flagellaless flaABΔdnaJ mutant being the most affected strain, and the NS4 transposant exhibiting an intermediate phenotype between the deletion mutant and the wild-type. Besides DnaJ, additional chaperones have been shown to be crucial for effective symbiosis with legumes. Compatible with our results, chaperone DnaK is also required in *S. meliloti* for efficient symbiosis [71]. GroEL proteins, which in *E. coli* are responsible for the correct folding of 10 to 15% of the cellular proteins [72], also play fundamental roles in infection and nitrogen fixation [73,74,75]. During the symbiotic interaction, rhizobia face several stresses outside and inside the plant host, including osmotic stress, low pH, low oxygen, or the presence of plant-produced reactive oxygen species and antimicrobials (flavonoids and nodule-specific cysteine-rich (NCR) peptides) [76,77]. Interestingly, the expression of *dnaJ* has been shown to be upregulated upon in vitro exposure to two cationic NCR peptides [78], suggesting a role for the chaperone in the bacterial response to peptides that, in planta, govern the differentiation of rhizobia. The role of chaperones such as DnaJ in the establishment of an efficient nitrogen-fixing symbiosis might be connected to its role in bacterial adaptation to stress. However, a signaling symbiotic role cannot be discarded. In *Salmonella*, DnaJ not only helps to counteract the oxidative stress generated inside macrophages but also participates in the activation of pathogenicity-related genes [51]. It would be interesting to unveil whether such a specific role in the interaction with legumes can also be associated to rhizobial DnaJ proteins.

## 4. Materials and Methods

### 4.1. Bacterial Strains, Plasmids, and Growth Conditions

Bacterial strains and plasmids used in this study are listed in Appendix A. *Escherichia coli* strains were grown in Luria–Bertani (LB) medium [79]) at 37 °C; *S. meliloti* strains were grown at 28 ºC in tryptone–yeast extract (TY) medium [80] or in minimal medium (MM) [81]. When required, antibiotics were added at final concentrations of 200 μg·mL^−1^ ampicillin, 50 μg·mL^−1^ kanamycin, and 10 μg·mL^−1^ tetracycline for *E. coli*, and 200 μg·mL^−1^ kanamycin, 10 μg·mL^−1^ tetracycline, and 75 μg·mL^−1^ hygromycin for *S. meliloti*. All antibiotics and reagents were obtained from Sigma-Aldrich (Steinheim, Germany), unless otherwise specified.

### 4.2. Isolation of GR4flaAB Mutants Insensitive to 2-TDC

Transposon Tn*5* mutagenesis of *S. meliloti* GR4flaAB was carried out as previously described [82] using strain S17-1 carrying pSUP2021 as the donor. After selection on MM plates supplemented with kanamycin, individual transposants were assayed for surface motility in the presence of volatile 2-TDC, as described below (Section 4.4). Those clones showing impaired surface motility in response to the volatile in three independent experiments were considered insensitive to 2-TDC and were selected for further characterization. The point of insertion for Tn*5* mutants was determined by arbitrary PCR, as described previously [83], using the primers specified in Appendix A.

### 4.3. Construction of Plasmids and S. meliloti Strains

Primers used to obtain the different plasmid constructs and *S. meliloti* mutant strains are listed in Appendix A. To obtain plasmid pJ-dnaJ used in genetic complementation experiments, a 1241-bp fragment containing the *S. meliloti* GR4 *dnaJ* gene was amplified from genomic DNA using primers dnaJ-F and dnaJ-R. The resulting PCR product was first cloned into pCR2.1-TOPO and sequenced, and then subcloned as a *Hin*dIII fragment into pJB3Tc19 [84] keeping *dnaJ* transcription in the same orientation as *lacZ*. Mutant strains GΔdnaJ and flaABΔdnaJ were obtained by allelic replacement of their wild-type *dnaJ* gene with a markerless in-frame deletion version of the locus. The deleted version of *dnaJ* was generated by overlap extension PCR using primers dnaJ-1 to dnaJ-3 and dnaJ-R. The resulting 1126-bp fusion product, in which a deletion of 768-bp was created in the coding sequence of *dnaJ*, was cloned into pCR2.1-TOPO and sequenced. Then, the insert was subcloned into the suicide vector pK18*mobsacB* as an *Eco*RI-*Hin*dIII fragment. The resulting plasmid pK18-ΔdnaJ was mobilized to GR4 or GR4flaAB by biparental matings using S17-1, and double cross-over events were selected, as previously described [85]. PCR amplifications were performed with the proofreading Phusion high-fidelity DNA polymerase (Thermo Scientific, Waltham, Massachusetts, USA). Deletion mutant strains were checked by Southern hybridization using a specific probe.

### 4.4. Motility Assays

Swimming motility was examined on Bromfield medium (BM) (0.04% tryptone, 0.01% yeast extract, and 0.01%CaCl_2_·2H_2_O) containing 0.3% Bacto agar after inoculation of 3-µL droplets of rhizobial cultures grown in TY broth (O.D._600nm_ = 1). Surface motility was assayed, as previously described [30,36]. Briefly, *S. meliloti* cells grown in TY broth to late logarithmic phase (O.D._600nm_ = 1–1.2) were pelleted, washed twice in MM, and resuspended in 0.1 volume of the latter medium. Two μL aliquots of this bacterial suspension (ca. 2 × 10^7^ cells) were dispensed and allowed to dry for 10 min onto the surface of plates containing 20 mL of semisolid MM containing 0.6% or 1% Noble Agar Difco (BD, Le Pont de Claix, France), which were previously air-dried at room temperature for 15 min. MM (0.6%) was used to test the surface motility of bacterial strains in the absence of 2-TDC, whereas MM (1%) was used to evaluate the response of *S. meliloti* strains to volatile 2-TDC. For assays in the presence of volatile 2-TDC, 20 µL of a 50 mM 2-TDC solution prepared in ethanol were applied onto the lid of the plate just before sealing with parafilm and incubation face-down. For control treatments, the same volume of pure ethanol was applied to the plates.

### 4.5. Stress Tolerance Assays

*S. meliloti* cells grown in TY broth to an OD_600nm_ of 0.5–0.7 were pelleted, washed twice in MM, and resuspended in the same volume of the latter medium. These bacterial suspensions were serially diluted and 10 µL of each dilution were spotted on solid MM plates supplemented with 300 mM NaCl (osmotic stress) or with different concentrations of H_2_O_2_ (oxidative stress), as indicated. To determine cell survival after exposure to H_2_O_2_, *S. meliloti* cultures were grown in MM broth up to an OD_600nm_ of 0.5–0.7 and then diluted 1:100 in MM. These bacterial suspensions were split in half and H_2_O_2_ was applied to one-half of the cultures to a final concentration of 1 mM. The reaction mixtures were incubated for 90 min at 30 °C. The survival rate was calculated by determining the colony-forming units (CFU) obtained on TY after plating serial dilutions of nontreated cultures and cultures exposed to H_2_O_2_. These assays were performed with three replicates per experiment.

### 4.6. Plant Assays

Alfalfa seeds (*Medicago sativa* L. cv. Victoria) were surface-sterilized and germinated, as previously described [86]. For nodulation kinetics/infectivity tests, alfalfa seedlings were grown in hydroponic cultures under axenic conditions in glass tubes containing nitrogen-free nutrient solution (one plant per tube) [86]. Ten-day-old plants (a total of 20–24 replicates) were inoculated with 1 mL of a rhizobial suspension containing 5 × 10^6^ cells. Prior to this inoculation, bacteria were grown to the exponential phase (OD_600_ = 0.5–0.6) in TY broth and diluted 100-fold in sterile water. After inoculation, the number of nodules per plant was recorded daily.

To obtain plant material for nitrogenase activity (see Section 4.7) and nodules for microscopy (Section 4.8), alfalfa plants were grown in Leonard jars, as previously described [30]. In this case, ten seedlings per jar were placed equidistantly from the center and immediately inoculated by applying 5 mL of a rhizobial suspension containing 1×10^3^ cells/mL to the center of the jar. The appearance and dry weight of the aerial part of plants, morphology, and the number of nodules, as well as nodule fresh weight and ARA were determined one month after inoculation.

### 4.7. Acetylene Reduction Activity (ARA)

Nitrogenase activity was determined by acetylene reduction, as described previously [87]. Four nodulated alfalfa roots were incubated at room temperature in vials containing C_2_H_2_ (10%, vol/vol) in air and sealed with serum caps. Aliquots of 0.1 mL were taken after 1 h of incubation and analyzed for ethylene in a Hewlett Packard 5890 gas chromatograph (Spring, TX, USA) equipped with a Poropak R column.

### 4.8. Microscopy Studies

Bacterial cells were observed using transmission electron microscopy (TEM). Cells were obtained from colonies grown on 1% agar MM plates. Carbon-coated Formvar grids were placed for 5 min on top of a drop of water previously applied to the colony. The grids were then washed twice in water for 1 min and stained with 2% (wt/vol) uranyl acetate for 3 min. The grids were allowed to air-dry for at least 1 h and visualized using a JEOL JEM-1011 transmission electron microscope with a 100-kV beam at the Microscopy Service of the Estación Experimental del Zaidín, Granada, Spain. Images were captured using an Orius Gatan charge-coupled-device camera.

Optical microscopy studies of alfalfa nodules were performed, as described previously [88]. Briefly, entire and half sections of nodules (30 days postinoculation) were immediately fixed in 4% (*v*/*v*) glutaraldehyde in 0.1 M cacodylate buffer, pH 7.2 for 1 h under vacuum conditions overnight at 4 °C. Samples were washed several times in 0.1 M cacodylate buffer, pH 7.2, dehydrated in acetone at progressively higher concentrations and embedded in Spurr resin (low viscosity embedding kit, by Dr. Spurr). The semithin sections (0.6–1 µm) were obtained on a Leica EM UC7 ultramicrotome, stained with Toluidine blue, and viewed in an Olympus BX61 light microscope. The number of infected versus noninfected cells was determined using transverse sections taken from the middle region of at least three different nodules induced by each strain.

### 4.9. Bioinformatic Analysis

Domain architecture of *S. meliloti* DnaJ was obtained using InterPro [89]. Multiple sequence alignment of DnaJ proteins was performed using MUSCLE hosted by EMBL-EBI [90].

## Figures and Tables

**Figure 1 ijms-24-05848-f001:**
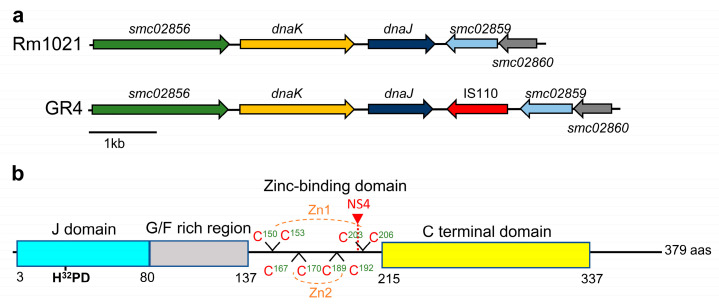
In silico analyses of *S. meliloti dnaJ*. (**a**) Genomic context of *dnaJ* in *S. meliloti* strains Rm1021 and GR4. The arrows represent open reading frames. The transposase encoded by the insertion sequence IS110 in GR4 is shown in red. *smc02856* encodes a putative penicillin-binding protein; *dnaK* and *dnaJ* code for the Hsp70 and Hsp40 chaperones, respectively; *smc02859*-*smc02860* encodes hypothetical proteins of unknown function. (**b**) Domain architecture of the *S. meliloti* DnaJ protein. The conserved HPD motif in the J-domain is shown in bold. The eight cysteines putatively conforming two zinc-binding domains are indicated in red. The location of the Tn*5* insertion in the NS4 transposant is shown with a red triangle.

**Figure 2 ijms-24-05848-f002:**
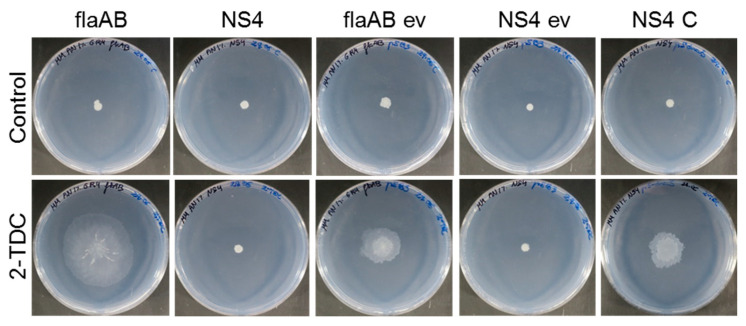
The inactivation of *dnaJ* in transposant NS4 is responsible for its insensitivity to 2-TDC. Surface motility assays on MM (1% agar) in the presence or absence of volatile 2-TDC. About 20 µL of either a solution containing 1 µmol 2-TDC or ethanol (Control) was applied to the lid of the plates just before incubation. flaAB, strain GR4flaAB; ev, empty vector pJB3; and C, complementing plasmid pJ-dnaJ. Representative pictures of the motilities exhibited after 48 h of incubation are shown.

**Figure 3 ijms-24-05848-f003:**
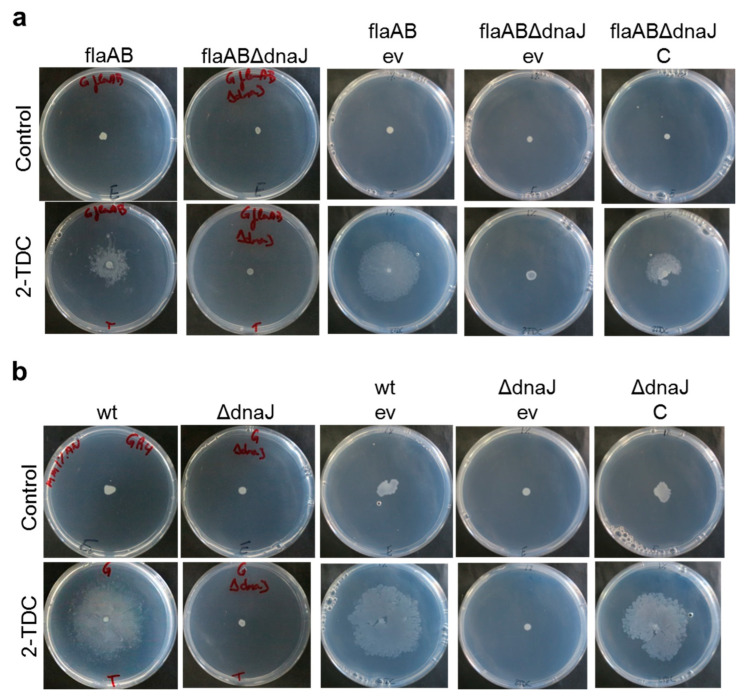
Surface motility exhibited by *S. meliloti dnaJ* deletion mutants and complemented strains. (**a**) GR4flaAB-derivative strains and (**b**) GR4-derivative strains were assayed on MM (1% agar) in the presence or absence of volatile 2-TDC. About 20 µL of either a solution containing 1 µmol 2-TDC or ethanol (Control) was applied to the lid of the plates just before incubation. flaAB, strain GR4flaAB; wt, strain GR4; ΔdnaJ, strain GΔdnaJ; ev, empty vector pJB3; and C, complementing plasmid pJ-dnaJ. Representative pictures of the motilities exhibited after 48 h of incubation are shown.

**Figure 4 ijms-24-05848-f004:**
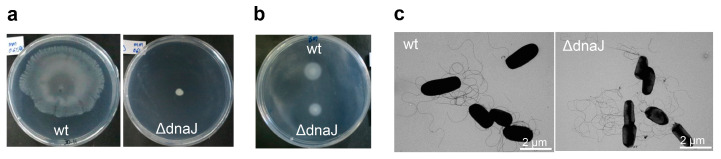
Swarming, swimming, and flagella production in the wild-type strain GR4 (wt) and its *dnaJ* deletion derivative mutant (ΔdnaJ). (**a**) Surface motility assays on semisolid MM (0.6 %). Representative pictures of the motility exhibited by each strain after 48 h of incubation at 28 °C are shown. (**b**) Swimming motility assay in BM (0.3%). Pictures were taken 72 h after inoculation. (**c**) Transmission electron microscopy images showing flagella production of wild-type and *dnaJ* mutant cells. Scale bars are indicated.

**Figure 5 ijms-24-05848-f005:**
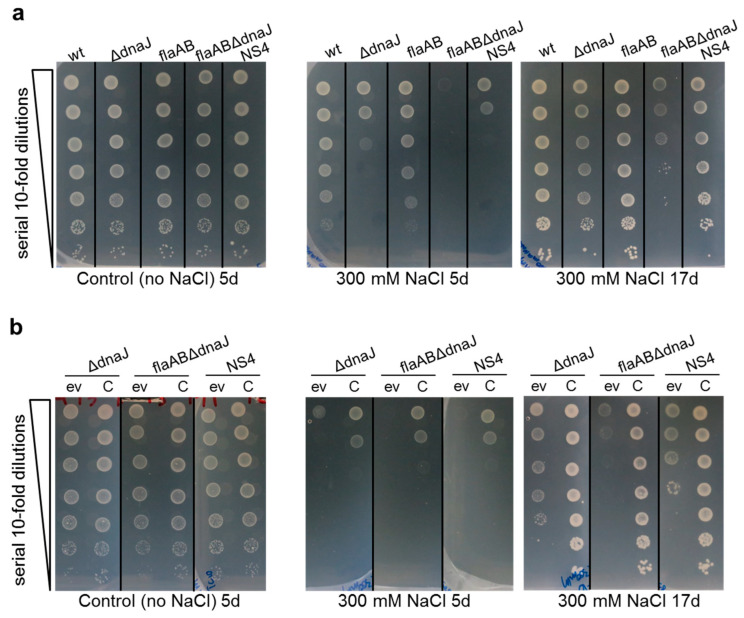
DnaJ loss-of-function in *S. meliloti* leads to increased sensitivity against salt stress. (**a**) Salt stress tolerance of GR4 (wt) and its *dnaJ* deletion mutant (ΔdnaJ), the flagellaless GR4flaAB (flaAB) and its *dnaJ* deletion (flaABΔdnaJ), and Tn*5* insertion (NS4) mutants. (**b**) Salt stress tolerance of *dnaJ* mutants carrying the empty vector (ev) or the *dnaJ*-expressing plasmid construct (C). Cell suspensions in MM were prepared from mid-log cultures at the same OD_600_, and 10 µL of 10-fold serial dilutions were spotted on MM agar plates supplemented or not with 300 mM NaCl. Plates were incubated at 28 °C for 5 (5 d) or 17 days (17 d).

**Figure 6 ijms-24-05848-f006:**
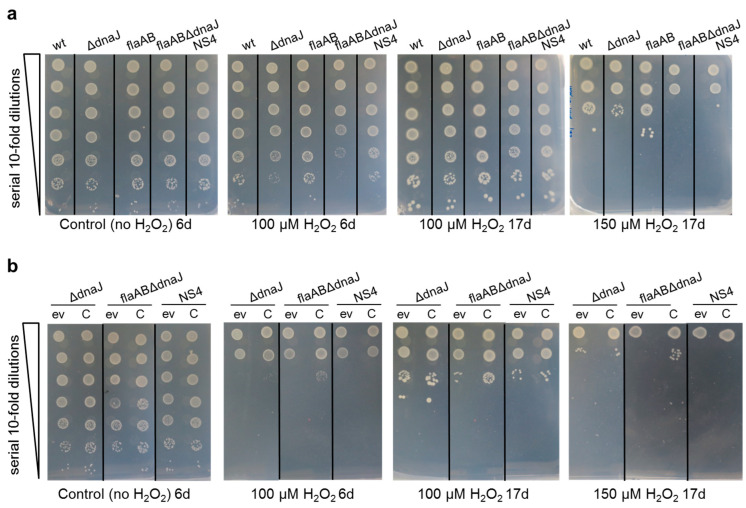
DnaJ loss-of-function in *S. meliloti* leads to increased sensitivity against oxidative stress. (**a**) Oxidative stress tolerance of GR4 (wt) and its *dnaJ* deletion mutant (ΔdnaJ), the flagellaless GR4flaAB (flaAB) and its *dnaJ* deletion (flaABΔdnaJ), and Tn*5* insertion (NS4) mutants. (**b**) Oxidative stress tolerance of *dnaJ* mutants carrying the empty vector (ev) or the *dnaJ*-expressing plasmid construct (C). Cell suspensions in MM were prepared from mid-log cultures at the same OD_600_, and 10 µL of 10-fold serial dilutions were spotted on MM agar plates supplemented or not with different concentrations of H_2_O_2_. Plates were incubated at 28 °C for 6 (6 d) or 17 days (17 d).

**Figure 7 ijms-24-05848-f007:**
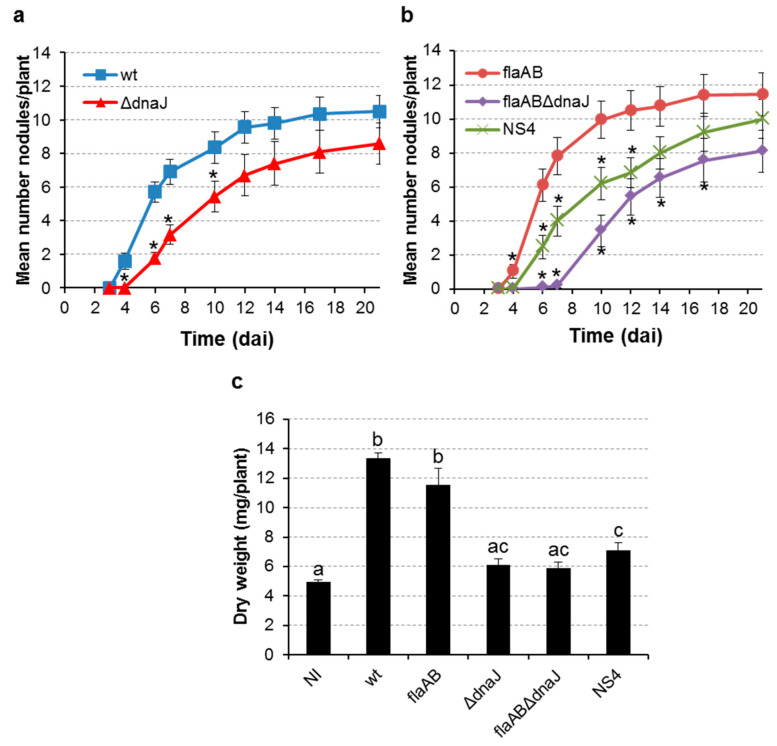
The symbiotic phenotype of *S. meliloti dnaJ* mutants on alfalfa plants. Nodulation kinetics of (**a**) GR4 strain (wt) and its derivative *dnaJ* deletion mutant (ΔdnaJ), and (**b**) the flagellaless strain GR4flaAB (flaAB) and its *dnaJ* deletion (flaABΔdnaJ), and transposon insertion (NS4) derivative mutants. Asterisks indicate significant differences compared to the parental strain according to an ANOVA test (*p* < 0.05). (**c**) Symbiotic efficiency of *S. meliloti* strains, as determined by the shoot dry weight of alfalfa plants 21 days after inoculation (dai). NI, noninoculated plants. Different letters indicate significant differences according to an analysis-of-variance (ANOVA) test (*p* < 0.05).

**Figure 8 ijms-24-05848-f008:**
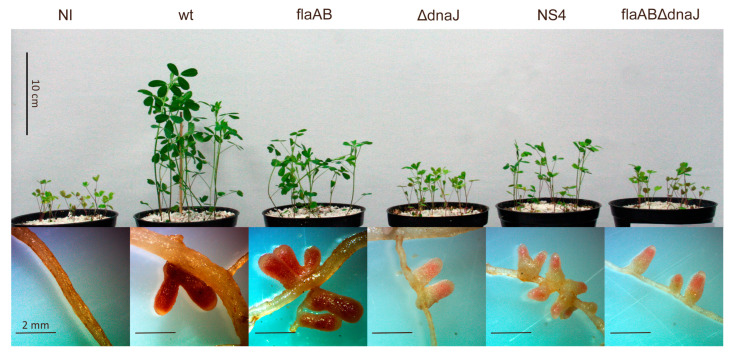
Plant growth and nodules formed by alfalfa inoculated with *S. meliloti dnaJ* mutants.

**Figure 9 ijms-24-05848-f009:**
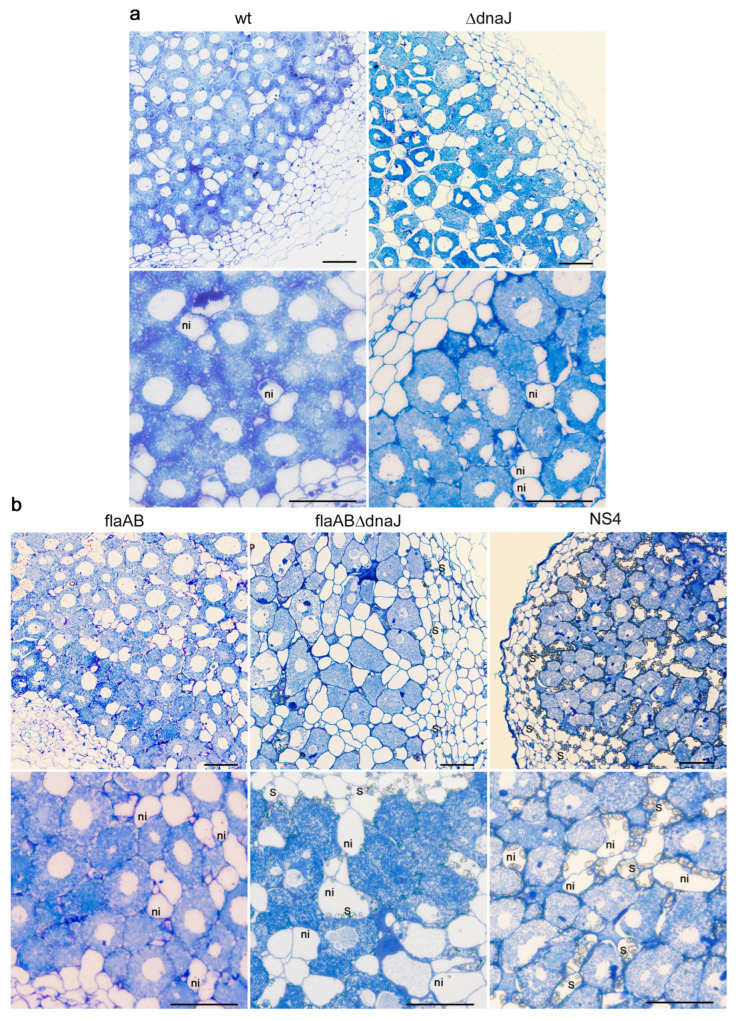
Light micrographs of sections of alfalfa nodules elicited by (**a**) GR4 strain (wt) and its derivative *dnaJ* deletion mutant (ΔdnaJ), and (**b**) GR4flaAB (flaAB), its *dnaJ* deletion mutant (flaAB ΔdnaJ), and the NS4 transposant. Bars correspond to 50 µm. ni, noninfected cells; s, starch granules.

**Table 1 ijms-24-05848-t001:** GR4flaAB-derived mutants insensitive to volatile 2-TDC.

Transposant	Tn*5* Location ^1^/Affected Locus GR4/Rm1021	Description of Gene Product
NS1	C770_GR4Chr0066*/smc02585* (*actS*)	Two-component sensor histidine kinase
NS2	C770_GR4Chr0263*/smc00334* (*cmK*)	Putative cytidylate kinase
NS3	C770_GR4Chr1253*/smc01800* (*ctaA*)	Putative heme A synthase
NS4	C770_GR4Chr0186/*smc02858* (*dnaJ*)	Probable chaperone protein
NS5	C770_GR4Chr3081/*smc03142*	Diguanylate cyclase (GGDEF) domain protein/Hypothetical transmembrane protein

^1^ The name of the affected locus in GR4 is given followed by the orthologous gene in Rm1021.

**Table 2 ijms-24-05848-t002:** Symbiotic parameters of alfalfa plants inoculated with *S. meliloti* wild-type, flaAB mutant, and *dnaJ* derivative mutant strains ^1^.

Strain	SDW ^2^	NN ^3^	NFW ^4^	ARA ^5^
wt	47.15 ± 4.38	11.46 ± 0.70	17.44 ± 1.58	32.76 ± 7.85
ΔdnaJ	13.76 ± 0.66 ***	11.47 ± 0.96	5.98 ± 0.68 ***	7.74 ± 0.64 ***
flaAB	24.61 ± 2.39	8.46 ± 1.16	9.64 ± 1.30	16.81 ± 1.71
flaABΔdnaJ	12.22 ± 0.78 ***	8.06 ± 0.84	4.63 ± 0.56 ***	7.03 ± 0.66 ***
NS4	14.3 ± 1.15 ***	11.31 ± 1.05 *	4.99 ± 0.46 ***	8.10 ± 1.35 **
NI ^6^	8.74 ± 0.71	-	-	-

^1^ Numbers are mean values (± standard error of the mean, SEM) per plant. Data were obtained from at least twenty-four plants inoculated with each rhizobial strain, 30 days after inoculation. Data obtained from plants inoculated with the *dnaJ* mutants were individually compared with the values obtained in plants inoculated with their parental strains, using the nonparametric test of Mann–Whitney. The presence of one or more asterisks denotes a significant difference with respect to data obtained in plants inoculated with the corresponding parental strain (* *p* < 0.05; ** *p* < 0.01; *** *p* < 0.001).^2^ Shoot dry weight in mg plant^−1^; ^3^ number of nodules per plant; ^4^ nodule fresh weight in mg plant^−1^; ^5^ acetylene reduction activity in nmol C_2_H_4_ plant^−1^ h^−1^; and ^6^ noninoculated.

## Data Availability

Not applicable.

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
