# Peer review of "Sinorhizobium meliloti* DnaJ Is Required for Surface Motility, Stress Tolerance, and for Efficient Nodulation and Symbiotic Nitrogen Fixation"

_ijms, 2023, doi:10.3390/ijms24065848_

Round 1

Reviewer 1 Report

The manuscript deals with the role of the DnaJ chaperone in free-living and symbiosis in Sinorhizobium meliloti (Sme GR4), an alphaprteobacterium capable of establishing symbiosis in Medicago species and fixing atmospheric nitrogen. The authors found several transposon-derived strains that were unable to sense 2-tridecanone as a signal molecule, which interfered with the surface motility of Sme. In addition, the authors test several specific mutant strains, including a clean one in dnaJ, testing the effects of this gene not only on 2-TD sensing, but also on other abiotic stresses and symbiotic performance, showing the pleiotropic nature of this chaperone.

The manuscript is well written, very clear in showing the results and describes extensively the experiments performed. Some questions arise from the hypothesis tested:

1- Clearly, dnaJ expression depends on the detection of 2-TD, but it also depends on other factors, as seen by the authors in the experiments presented. In any case, did the authors explore the promoter region of this operon? Is there only one, or more than one promoter predicted for dnaJ transcription? According to Barnett, M. J.et al. (2012. Dual RpoH sigma factors and transcriptional plasticity in a symbiotic bacterium. J Bacteriol 194:4983-4994), dnaK transciption is dependent on RpoH. It is important to look at this region, as it is apparently regulated not only by its own promoter, but also by transcoded small RNAs (Gao M, et al. Role of the Sinorhizobium meliloti global regulator Hfq in gene regulation and symbiosis. Mol Plant Microbe Interact. 2010 Apr;23(4):355-365. doi: 10.1094/MPMI-23-4-0355. PMID: 20192823; PMCID: PMC4827774). Even in silico, this section could be improved to clarify in depth the genomic context of dnaJ and its regulation (probably the presence of SMc02856 in Fig1 could be confusing).

2- Being dnaJ involved in surface motility, is there any clue about the attachment location to rhizoplane and the infectivity? Has been observed any positional different in nodule development in the dnaJ mutant strain?

3-The photos seem to demonstrate that these strains are not able to evolve to mature areas of the nodule. Please refer to Tiricz H, et al. (Antimicrobial nodule-specific cysteine-rich peptides induce membrane depolarization-associated changes in the transcriptome of Sinorhizobium meliloti. Appl Environ Microbiol. 2013 Nov;79(21):6737-46. doi: 10.1128/AEM.01791-13.) that showed that dnaJ nodule expression is under the action of NCRs (247 and 335), and thus improve the discussion section.

Minor remarks:

- the number of nodules should be logarithmically transformed prior to ANOVA testing.

Author Response

We would like to thank the reviewer for the helpful and constructive comments which without a doubt have helped us to improve the manuscript. Please find below a point-by-point response to the comments made. Page and line numbers refer to the revised version in which all the track changes are shown.

Point 1: Clearly, dnaJ expression depends on the detection of 2-TD, but it also depends on other factors, as seen by the authors in the experiments presented. In any case, did the authors explore the promoter region of this operon? Is there only one, or more than one promoter predicted for dnaJ transcription? According to Barnett, M. J.et al. (2012. Dual RpoH sigma factors and transcriptional plasticity in a symbiotic bacterium. J Bacteriol 194:4983-4994), dnaK transciption is dependent on RpoH. It is important to look at this region, as it is apparently regulated not only by its own promoter, but also by transcoded small RNAs (Gao M, et al. Role of the Sinorhizobium meliloti global regulator Hfq in gene regulation and symbiosis. Mol Plant Microbe Interact. 2010 Apr;23(4):355-365. doi: 10.1094/MPMI-23-4-0355. PMID: 20192823; PMCID: PMC4827774). Even in silico, this section could be improved to clarify in depth the genomic context of dnaJ and its regulation (probably the presence of SMc02856 in Fig1 could be confusing).

Response 1: Investigating the transcriptional regulation of dnaKdnaJ genes was not the aim of this study, which was mainly focused on the functional characterization of dnaJ in the free-living and symbiotic lifestyles of S. meliloti. However, following the reviewer’s suggestion, in the revised version of the manuscript we have included some additional information about the genomic context of dnaJ and on its regulation (page 4, lines 161-163; page 15, lines 517-525). As indicated in the manuscript (page 15, lines 517-525), expression of dnaJ might be rather complex. According to the global mapping of transcription start sites (TSSs) performed in the study by Schlüter et al (2013) (Schlüter et al., 2013 Global mapping of transcription start sites and promoter motifs in the symbiotic alpha-proteobacterium Sinorhizobium meliloti 1021. BMC Genomics, 14, 156), 2 TSSs have been identified upstream of the start codon of dnaK, and 4 TSSs upstream of the start codon of dnaJ, suggesting that dnaJ can also be expressed independently of dnaK. In addition, the antisense RNA SMc_asRNA_774 that overlaps the 5’-UTR of the mRNA initiated upstream of dnaK was also identified. A summary of this information, together with the RpoH1-dependent regulation of the dnaKdnaJ operon has been included in the manuscript (page 15, lines 517-525).

Concerning the presence of smc02856 in Figure 1, we decided to keep it in the figure to make it clear that the grpE gene coding for the third component of the chaperone folding system DnaK/DnaJ/GrpE is not in the proximity of the dnaKdnaJ operon.

Point 2: Being dnaJ involved in surface motility, is there any clue about the attachment location to rhizoplane and the infectivity? Has been observed any positional different in nodule development in the dnaJ mutant strain?

Response 2: Under our experimental conditions, no obvious differences were observed concerning the position of nodules induced by the wild-type or the dnaJ mutant strains.

Point 3: The photos seem to demonstrate that these strains are not able to evolve to mature areas of the nodule. Please refer to Tiricz H, et al. (Antimicrobial nodule-specific cysteine-rich peptides induce membrane depolarization-associated changes in the transcriptome of Sinorhizobium meliloti. Appl Environ Microbiol. 2013 Nov;79(21):6737-46. doi: 10.1128/AEM.01791-13.) that showed that dnaJ nodule expression is under the action of NCRs (247 and 335), and thus improve the discussion section.

Response 3: As indicated by Reviewer 3, the macroscopic appearance of the nodules induced by dnaJ mutants are suggestive of early senescence, a concept that we have included in the revised version of the manuscript (page 11, lines 418-421). The up-regulation of dnaJ by NCR peptides is an interesting result that suggests that DnaJ might be required for counteracting the effects caused by the plant-produced NCR peptides and therefore allow for an efficient symbiosis. This concept together with the reference Tiricz et al. (2013) have been included to improve the Discussion section (page 16, lines 607-610). We would like to thank the reviewer for this valuable suggestion.

Point 4: Minor remarks:

- the number of nodules should be logarithmically transformed prior to ANOVA testing.

Response 4: The transformation suggested by the reviewer cannot be applied to our data because at the beginning of the nodulation kinetics, many plants have 0 nodules.

Reviewer 2 Report

The paper by Brito-Santana et al. reports the description of pleiotropic phenotypes of dnaJ mutants of S. meliloti, particularly in connection with motility, stress response and symbiotic nitrogen fixation.

It is an interesting work, with a solid experimental design and results clearly presented. Although the role of DnaJ in a numerous cellular process is well known and it is also known to affect rhizobial symbiosis, the paper is nevertheless original as the function of this protein in S. meliloti is little recognized. In addition, a relevant role of DnaJ on surface translocation is distinctly demonstrated.

Following a list of comments to be address:

Major criticisms:

-         Data on oxidative stress should be better considered. Looking at results in fig. 6, the differences between wt and dnaJ strains is really small as also stated by the authors, the same is true for the difference between dnaJ mutants and their respective complemented strains. In the absence of a strong statistical analysis (data on cell survival not reported) these differences are not convincing and should be cautiously considered.  I suggest not to focus too much on this aspect in the discussion and particularly, in the title, add “salt” to stress as this is what is clearly demonstrated.

-       The title does not mention surface motility, that is misleading as this is the main phenotype used for the selection and first characterization of the mutant.

Minor comments

-       Regarding fig. S1, it seems there is a lot of variation in the motility of the parental strain GR4flaAB, though it is clear that the mutants do not move, it seems that accidental factors could affect the outcome of the test.

-       Three NS mutants were affected in genes coding for known proteins, could the authors comment on the decision to start with DnaJ?

-       Tn induced mutant NS4 shows a phenotype clearly different from the dnaJ deleted mutant, possibly because of a residual peptide is produced. Could the authors verify what of dnaJ is actually transcribed and translated? On the same line, the additional growth after 17 days of the dnaJ mutants in high salt medium should be tested/discussed for the possibility that new mutations/loss of transposon were selected for under stress conditions.

Author Response

We would like to thank the reviewer for the helpful and constructive comments which without a doubt have helped us to improve the manuscript. Please find below a point-by-point response to the comments made. Page and line numbers refer to the revised version in which all the track changes are shown.

MAJOR CRITICISMS

Point 1: Data on oxidative stress should be better considered. Looking at results in fig. 6, the differences between wt and dnaJ strains is really small as also stated by the authors, the same is true for the difference between dnaJ mutants and their respective complemented strains. In the absence of a strong statistical analysis (data on cell survival not reported) these differences are not convincing and should be cautiously considered.  I suggest not to focus too much on this aspect in the discussion and particularly, in the title, add “salt” to stress as this is what is clearly demonstrated.

Response 1: Following the reviewer’s suggestion, we have reduced the part of the discussion dealing with the participation of DnaJ in oxidative stress tolerance (page 16, lines 580-589). However, we disagree with the suggestion of adding “salt” to stress in the title because this would be too restrictive considering the results obtained in this study. Although not as striking as what we showed for salt stress, we believe that the delay in growth exhibited by the three dnaJ mutant strains in the presence of 100 µM H2O2 together with the decrease in cell survival of at least one order of magnitude shown by the GR4flaAB derivative dnaJ mutants in the presence of 150 µM H2O2 (Figure 6a), are sufficient to indicate a role for DnaJ in the adaptation to oxidative stress. The lack of “convincing” complementation could be due to the combined effects of H2O2 and the antibiotic tetracycline on DnaJ activity. Besides the effect caused by H2O2, and as shown in Figures 5b and 6b, the presence of the antibiotic tetracycline and/or the burden of carrying additional DNA, impairs even more the mutant’s growth in media containing high salt or H2O2, indicating that DnaJ participates in tolerance to additional stresses (page 8, lines 311-313; page 10, lines 361-364). Moreover, and as described in the Discussion section (page 14, lines 493-502), our genetic and transcriptomic data suggest that 2-TDC is triggering some as yet unknown stress-related response that does not affect bacterial growth (please see also the answer to the reviewer’s comment #4). The inability of dnaJ mutants to adapt to such a stress could explain their insensitivity to the volatile. For all these reasons, and considering also the reviewer’s comment #2, the title has been changed to: Sinorhizobium meliloti DnaJ is required for surface motility, stress tolerance, and for efficient nodulation and symbiotic nitrogen fixation.

Point 2: The title does not mention surface motility, that is misleading as this is the main phenotype used for the selection and first characterization of the mutant.

Response 2: Indeed, therefore to avoid any confusion the title has been changed in order to mention the role of DnaJ in surface motility. As mentioned in Point 1, the new title is: Sinorhizobium meliloti DnaJ is required for surface motility, stress tolerance, and for efficient nodulation and symbiotic nitrogen fixation.

MINOR COMMENTS

Point 3: Regarding fig. S1, it seems there is a lot of variation in the motility of the parental strain GR4flaAB, though it is clear that the mutants do not move, it seems that accidental factors could affect the outcome of the test.

Response 3: Surface motility assays are very variable, even when the experimental conditions are strictly controlled to avoid accidental factors. For this reason, many repetitions in which the corresponding controls must always be included, are performed before reaching conclusions. For more information related to the challenging nature of surface motility assays in bacteria and especially in rhizobia, the reviewer is referred to our publication Bernabéu-Roda, L.M., López-Ráez, J.A., and Soto, M.J. Analyzing the effect of strigolactones on the motility behavior of Rhizobia. En Prandi, C., Cardinale, F. (eds), Strigolactones. Methods in Molecular Biology, vol. 2309, Humana, New York, NY. (2021) pp 91-103. https://doi.org/10.1007/978-1-0716-1429-7_8.

Point 4: Three NS mutants were affected in genes coding for known proteins, could the authors comment on the decision to start with DnaJ?

Response 4: As indicated in the manuscript, stress-related genes were found up-regulated in response to 2-TDC in a previous transcriptome analyses (lines 498-500). This led to our hypothesis that 2-TDC is triggering some kind of stress-related response. Therefore, we focused on DnaJ as it is known to be involved in stress tolerance in different bacteria. Of course, we are also interested in the remaining three proteins and studies are underway.

Point 5: Tn induced mutant NS4 shows a phenotype clearly different from the dnaJ deleted mutant, possibly because of a residual peptide is produced. Could the authors verify what of dnaJ is actually transcribed and translated? On the same line, the additional growth after 17 days of the dnaJ mutants in high salt medium should be tested/discussed for the possibility that new mutations/loss of transposon were selected for under stress conditions.

Response 5: The reviewer is right in that we cannot rule out the possibility that the greater tolerance exhibited by the NS4 transposant is the result of loss of transposon or the selection of suppressor mutations. This possibility is now mentioned in the Discussion (lines 578-580). We do not have data to verify what of dnaJ is actually transcribed or translated in NS4. This would be interesting for future studies aimed at advancing in the functional characterization of DnaJ at the molecular level.

Reviewer 3 Report

The paper by Brito-Santana et al., describes the role of the DnaJ gene in adaptation to stress and during symbiosis.

This paper is nicely written, the work well designed and it provides a lot of biological indications in VOC perception and VOC roles in Rhizobium adaptation to environment including symbiosis.

The aim of the work was to find actors in VOC (2-Tridecanone, 2-TDC) perception in Sinorhizobium. Using a Tn5 mutant approach the authors identified 5 genes. When mutated they made the bacteria insensitive to 2-TDC.  One of them was DnaJ. They reconstructed a mutant by deleting the DnaJ sequence in the WT strain or in a Flagellin mutant. The single and double mutants were tested for swarming and resistance to stress. These results are clear as they have been able to complement the different mutant phenotypes with a copy of the DnaJ gene.

Below are my comments:

Second sentence of the introduction: It is not correct to describe nodules as a tumor-like organs. They are organs. Tumor means non-organized growing tissue, and the nodules are highly organized.

The authors describe a survival experiment in presence of a high (1 mM) amount of H2O2. They indicate that the percentage of survival cells was significantly affected by the flAB deletion but numbers are not given. Please add some quantified data in Sup Material.

Similarly, there is a sentence in the Mat and Met section stating: “The survival rate was calculated by determining the colony forming units (CFU) obtained on TY after plating serial dilutions of non-treated cultures and cultures exposed to H2O2.” There is no data (survival rate) corresponding to this. It might be related with the point above.

For the nodulation experiments, the phenotypes of the mutant nodules indicate an early senescence phenotype. The mutant nodules are pink in the apical part and probably greenish in the basal part indicating senescence (leghemoglobin degradation). Please indicate it in the corresponding result part instead of “the dnaJ mutants were smaller and presented a lighter pink color, which was rather concentrated on the apical part of the organ (Figure 8).”

In order to better visualize this early senescence, the authors should have shown entire section of the nodules (from root to the meristematic region). This would have better illustrated the difference between the WT and the DnaJ mutants.

In order to quantify the reduced infection, the authors should also quantify the infected versus non infected cell in the nodule sections, taking care to do the measurement at the same position (early zone III) of the nodule. This could be a nice result.

In this work, the in vitro phenotypes are nicely correlated with the presence of 2-TDC as the phenotypes are not observed in absence of this compound. In contrast the nodulation experiment just indicate that DnaJ is necessary but this can be independent of the 2-TDC. The authors should discuss the results of their previous paper (Environ Microbiol. 2018 20(6):2049-2065. doi: 10.1111/1462-2920.14083.), showing that 2-TDC can delay and reduce nodulation, suggesting that the DnaJ phenotype is related to 2-TDC production. However, the phenotype of the nodules formed in presence of 2-TDC is not indicated in the previous paper. May be this should also be discussed.

Author Response

We would like to thank the reviewer for the helpful and constructive comments which without a doubt have helped us to improve the manuscript. Please find below a point-by-point response to the comments made. Page and line numbers refer to the revised version in which all the track changes are shown.

Point 1: Second sentence of the introduction: It is not correct to describe nodules as a tumor-like organs. They are organs. Tumor means non-organized growing tissue, and the nodules are highly organized.

Response 1: The expression “new organs” has been now used instead of “tumor-like organs” (page 1, line 38).

Point 2: The authors describe a survival experiment in presence of a high (1 mM) amount of H2O2. They indicate that the percentage of survival cells was significantly affected by the flAB deletion but numbers are not given. Please add some quantified data in Sup Material.

Response 2: A new Figure (Figure S4) showing the effect of H2O2 on S. meliloti GR4 and GR4flaAB cell survival after exposure to 1 mM H2O2 for 90 min has been included in the revised Supplementary Material, and mentioned in the main manuscript on page 10 line 356.

Point 3: Similarly, there is a sentence in the Mat and Met section stating: “The survival rate was calculated by determining the colony forming units (CFU) obtained on TY after plating serial dilutions of non-treated cultures and cultures exposed to H2O2.” There is no data (survival rate) corresponding to this. It might be related with the point above.

Response 3: Please see Response 2.

Point 4: For the nodulation experiments, the phenotypes of the mutant nodules indicate an early senescence phenotype. The mutant nodules are pink in the apical part and probably greenish in the basal part indicating senescence (leghemoglobin degradation). Please indicate it in the corresponding result part instead of “the dnaJ mutants were smaller and presented a lighter pink color, which was rather concentrated on the apical part of the organ (Figure 8).”

Response 4: We thank the reviewer for this important comment. The suggested change has been included in the Results section (page 11, lines 418-421).

Point 5: In order to better visualize this early senescence, the authors should have shown entire section of the nodules (from root to the meristematic region). This would have better illustrated the difference between the WT and the DnaJ mutants.

Response 5: The reviewer is absolutely right. Optical microscopy of longitudinal sections of nodules would show the reduction of the nitrogen fixation zone (zone III) together with the enlargement of the senescent zone (zone IV) in nodules induced by the mutants compared with those induced by the parental strains. However, in this study, our interest was mainly focused on testing infection of symbiotic cells in early zone III. To better characterize the premature aging of nodules induced by dnaJ mutants, additional experiments including the already mentioned optical microscopy of longitudinal sections as well as analyses of their microstructure by transmission electron microscopy, and analyses of molecular markers of nodule senescence would be desirable but these studies were beyond the scope of our study.

Point 6: In order to quantify the reduced infection, the authors should also quantify the infected versus non infected cell in the nodule sections, taking care to do the measurement at the same position (early zone III) of the nodule. This could be a nice result.

Response 6: Following the reviewer’s suggestion, the number of infected cells versus non-infected cells has been determined using transverse sections taken from the middle region of nodules induced by the different strains. The results and the corresponding statistical analyses show a greater number of non-infected cells in nodules induced by the dnaJ mutants compared to nodules induced by the parental strains. These data have been included in pages 13-14, lines 458-468. In addition, a sentence indicating how these analyses were done has been included in the Materials and Methods section (page 18-19, lines 727-729).

Point 7: In this work, the in vitro phenotypes are nicely correlated with the presence of 2-TDC as the phenotypes are not observed in absence of this compound. In contrast the nodulation experiment just indicate that DnaJ is necessary but this can be independent of the 2-TDC. The authors should discuss the results of their previous paper (Environ Microbiol. 2018 20(6):2049-2065. doi: 10.1111/1462-2920.14083.), showing that 2-TDC can delay and reduce nodulation, suggesting that the DnaJ phenotype is related to 2-TDC production. However, the phenotype of the nodules formed in presence of 2-TDC is not indicated in the previous paper. May be this should also be discussed.

Response 7: Here, we believe that there is some misunderstanding. In this work, 2-TDC was used only for data shown in Figures 2 and 3, in which different S. meliloti strains were evaluated for the 2-TDC-induced surface spreading. However, the swarming, swimming and flagella production exhibited by GR4 and its dnaJ derivative mutant shown in Figure 4, the bacterial tolerance to salt and oxidative stress shown in Figures 5 and 6, and the nodulation experiments shown in Figures 7 and 8 were all performed in the absence of 2-TDC. The results presented in our study are compatible with the notion that S. meliloti dnaJ mutants show a phenotype different from the wild-type when they face some kind of stress. This obviously occurs when bacteria are exposed in vitro to high concentrations of salt or hydrogen peroxide and also in symbiosis because rhizobia probably encounter several stresses outside and inside the plant host, as we explained in the Discussion (page 16, lines 604-607). The dnaJ mutants are also unable to respond to 2-TDC (i.e. unable to spread over surfaces in the presence of 2-TDC) and we believe that this is because they are unable to adapt to some kind of volatile-induced stress, the nature of which is still unknown. This idea was described in the Discussion (lines 493-502). Moreover, DnaJ is also essential for the swarming motility exhibited by GR4 in the absence of 2-TDC (Figure 4a). Our hypothesis is that swarming motility in S. meliloti involves a stress response in which DnaJ plays a role. For more clarity, a new sentence highlighting this hypothesis has been included in the Discussion (lines 507-510).

Concerning the results of our previous paper, we do not have any evidence indicating that the delay and reduced nodulation exhibited by the dnaJ mutants is related to 2-TDC production. It is more likely that the similarity in phenotypes is a coincidence. We believe that the delay and reduced nodulation exhibited by the dnaJ mutants is probably due to the inability to adapt to stressful conditions (osmotic stress, acidic pH, flavonoids, etc) encountered in the rhizosphere/rhizoplane. With regard to the phenotype of the nodules formed in the presence of exogenously added 2-TDC in our previous paper (López-Lara et al. 2018. Environ. Microbiol. 20(6):2049-2065), they were indistinguishable from those formed in the absence of 2-TDC.

Round 2

Reviewer 2 Report

the authors' responses to my comments are convincing. I believe the manuscript can be published in its current form